# FDCSP Is an Immune-Associated Prognostic Biomarker in HPV-Positive Head and Neck Squamous Carcinoma

**DOI:** 10.3390/biom12101458

**Published:** 2022-10-12

**Authors:** Qingqing Wu, Tingru Shao, Guangzhao Huang, Zenan Zheng, Yingtong Jiang, Weisen Zeng, Xiaozhi Lv

**Affiliations:** 1Department of Oral and Maxillofacial Surgery, NanFang Hospital, Southern Medical University, Guangzhou 510515, China; 2Department of Oral and Maxillofacial Surgery, West China College of Stomatology, Sichuan University, Chengdu 610041, China; 3Department of Stomatology, Hexian Memorial Affiliated Hospital of Southern Medical University, Guangzhou 511400, China; 4Department of Cell Biology, School of Basic Medical Science, Southern Medical University, Guangzhou 510515, China

**Keywords:** FDCSP, HPV, head and neck squamous carcinoma (HNSC) TCGA, immune cell infiltration, CIBERSORT, CXCL13, TP53

## Abstract

**Background:** Head and neck squamous carcinoma (HNSC) poses a major threat to human life. The role of human papillomavirus (HPV) infection in the initiation and progression of HNSC is becoming more widely accepted. HPV-positive (HPV+) HNSC has shown unique responses to cancer therapies, which may be due to differences in immune cell infiltration. It is critical to determine how the immune responses to HPV in HNSC are regulated. **Methods:** Transcriptome data of HNSC from The Cancer Genome Atlas (TCGA) and the Gene Expression Omnibus (GEO) database were analyzed. Then, the CIBERSORT algorithm was used to calculate immune cell infiltration in HNSC. FDCSP expression level was detected by qPCR in the HNSC tissues collected from the Nanfang Hospital. **Results:** Follicular dendritic cell secreted protein (FDCSP) was highly expressed in HPV+ HNSC, and higher expression of FDSCP was associated with a favorable prognosis. In HPV+ HNSC samples, FDCSP significantly increased the proportion of T follicular helper cells (TFHs). FDCSP expression was also found to be associated with TP53 mutation status in HPV+ HNSC. The function of FDCSP was intimately connected to chemokine pathways, particularly with the C-X-C motif chemokine ligand 13 (CXCL13). We verified that the high expression of FDCSP in HPV+ HNSC and higher FDCSP is closely related to prognosis in HNSC samples we collected by qPCR. **Conclusions:** Collectively, these findings may provide fresh evidence that FDCSP is a potential chemokine-associated prognostic biomarker in HPV+ HNSC.

## 1. Introduction

Head and neck squamous carcinoma (HNSC) is the sixth most common cancer worldwide [1]. HNSC has multiple therapeutic etiological factors, including tobacco, alcohol consumption, mechanical stimulation, and HPV infection. Tobacco, alcohol, and betel quid intakes are thought to be responsible for a large proportion of HNSC, and despite the availability of effective vaccines against HPV, the proportion of HPV-related HNSC is increasing due to insufficient vaccine prevalence. Over the past few decades, it has been obvious that HPV-negative (HPV−) and HPV-positive (HPV+) HNSCs have distinct biochemical and clinical characteristics.

Human papillomavirus (HPV) is a tumor-associated virus with unenveloped double-stranded DNA (dsDNA). HPV infection causes nearly all cervical squamous cell carcinomas and approximately 40% of head and neck cancers [2,3]. In comparison to HPV− HNSC, HPV+ HNSC is a distinct disease entity with genetic, morphological, and clinical characteristics associated with an improved response to standard therapy and a favorable prognosis [4]. Some researchers have suggested treating HPV+ and HPV− HNSC separately and developing new treatment options for HPV-infected HNSC [5]. Despite significant progress in anti-PD-1 monoclonal antibody (mAb)-based immunotherapy, the overall survival (OS) of HNSC patients remains dismal [6,7,8,9,10]. Immunotherapy with anti-PD1 checkpoint inhibitors improves outcomes in patients with advanced HNSC regardless of HPV status [11]. Although a major step forward, first-generation anti-PD1 mAbs are nonspecific immunotherapies that do not take advantage of the tumor antigen specificity represented by HPV infection [8,9].

Because most previous studies compared HNSC or HPV+ groups to normal groups, studies comparing HPV+ against HPV− cancers are scarce. HPV infection has been shown to increase T-cell infiltration and activate immunological effector cells [12]. In addition, there is a strong correlation between increased T-cell inflammatory gene expression and HPV infection [12]. A recent study also found a strong B-cell signature in HPV+ HSNC by flow cytometry [13]. According to the research findings, HPV infection has a role in the tumorigenesis of HNSC and is related to the tumor microenvironment. Therefore, a comprehensive analysis of genes important for HPV+ HNSC and immune cell infiltration is highly desirable.

Follicular dendritic cell secreted protein (FDCSP, c4orf7) is a tiny, secreted protein that was discovered in follicular dendritic cells (FDCs). The FDCSP gene lies on chromosome 4q13 adjacent to clusters of proline-rich salivary peptides and C-X-C chemokines [14]. FDCSP is expressed mostly by activated FDCs and TNF-alpha-activated FDC-like cell lines. FDCSP specifically binds to activated B cells and functions as a regulator of antibody responses. It is also thought to contribute to tumor metastasis by promoting cancer cell migration and invasion. Unlike dendritic cells (DC), FDCs are not derived from bone-marrow hematopoietic stem cells but are of mesenchymal origin [15].

Our study performed immunological analyses using HNSC combined data from GEO and TCGA. We analyzed differentially expressed HPV genes and evaluated the correlation and prognosis of FDCSP with immune infiltration in both HPV+ and HPV− HNSC. These findings support the hypothesis that FDCSP is the immune-associated biomarker that regulates HPV-induced HSNC.

## 2. Materials and Methods

### 2.1. Microarray Data

This study retrieved microarray datasets from the Gene Expression Omnibus (GEO) database. The GSE65858 dataset [12], based on the Agilent GPL10558 platform, contained 74 samples of HPV+ and 196 samples of HPV− from Germany. The GSE117973 dataset, based on the Agilent GPL10558 platform, contained 23 samples of HPV+ and 54 samples of HPV−. The GSE55545 dataset, based on the Agilent GPL17077 platform, contained 8 samples of HPV+ and 16 samples of HPV−. The GSE40774 dataset, based on the Agilent GPL13497 platform, contained 58 HPV+ samples and 76 HPV− samples. The GSE39366 dataset, based on the Agilent GPL9053 platform, contained 14 samples of HPV+ and 124 samples of HPV−. The GSE6791 dataset, based on the Agilent GPL570 platform, contained 16 samples of HPV+ and 26 samples of HPV−. After combining multiple sets of data and adjusting for batch effects, differentially expressed genes (DEGs) were identified.

### 2.2. Identification of Differentially Expressed Genes

The DEGs were determined between HPV+ HNSC and HPV− HNSC samples with the “limma” R package. The |log2-fold change (FC)| > 2.5 and adj. *p*-value (FDR) < 0.05 were considered the cutoff criterion. The results of DEGs were introduced to the heatmap and the volcano map.

### 2.3. Functional and Pathway Enrichment Analyses of Differentially Expressed Genes

Gene Ontology (GO) enrichment analysis and Kyoto Encyclopedia of Genes and Genomes (KEGG) pathway analysis were conducted with the DEGs by adopting the “clusterProfiler” package [16]. GO enrichment analysis showed that the biological processes (BP) and molecular functions (MF) were correlated with DEGs. The biological pathways correlated with DEGs were revealed from KEGG pathway analysis. The threshold for enrichment significance was a *p*-value < 0.05.

### 2.4. Gene Set Enrichment Analysis

For gene set enrichment analysis (GSEA), we obtained the GSEA software (version 3.0) from the GSEA [17] (http://software.broadinstitute.org/gsea/index.jsp, accessed on 1 March 2022) website, according to which the FDCSP expression levels were divided into high expression groups (≥50%) and low expression groups (<50%) and were obtained from the Molecular Signatures Database [18] (http://www.gsea-msigdb.org/gsea/downloads.jsp, accessed on 1 March 2022). We downloaded the c5.go.bp, c5.go.mf, c5go, and KEGG symbol subsets to evaluate related pathways and molecular mechanisms, which were grouped based on gene expression profiles and phenotypes, and set the minimum gene set to 5. With a maximum gene set of 5000 and one thousand resamplings, a *p*-value of < 0.05 (as needed) and an FDR of < 0.25 (as needed) were considered statistically significant.

### 2.5. TIMER2.0 Database

TIMER2.0 (Tumor Immune Estimation Resource 2.0, http://timer.comp-genomics.org/, accessed on 1 March 2022) [19,20,21,22] is a comprehensive resource for the systematic analysis of immune infiltrates across diverse cancer types. In the outcome module, the Cox proportional hazard model covariates were overall survival and FDCSP expression, and the normalized coefficient of the infiltrate for each model across HPV status in HNSC cancer types is presented in a heatmap. Correlations between FDCSP expression and gene markers of tumor-infiltrating immune cells were explored via correlation modules by Spearman’s method. The gene expression level was displayed with Log2 RSEM. The FDCSP outcome in HNSC was analyzed in the Gene_Surv module. Timer 2.0 used the Cox proportional hazard model to evaluate the outcome significance of gene expression, which was optionally adjusted by clinical factors.

### 2.6. Immune Infiltration Analysis

To determine the association between the FDCSP expression level and immune infiltration according to TCGA tumor data, we used the Tumor Immune Estimation Resource 2.0 (TIMER2.0) (http://cistrome.org/TIMER/, accessed on 1 March 2022) web server and evaluated the correlation of the FDCSP expression level with the abundance of infiltrating immune cells in HNSC patients using the CIBERSORT_ABS algorithm. The *p*-values and partial correlation values were calculated by means of purity-adjusted Spearman’s rank correlation test.

### 2.7. Mutation Evaluation

We downloaded the Simple Nucleotide Variation dataset of level 4 of all TCGA samples processed by MuTect2 software [23] from GDC (https://portal.gdc.cancer.gov/, accessed on 1 March 2022), and we integrated the sample mutation data and gene expression data. We also filtered samples for synonymous mutations. A total of 493 HNSC samples with detected mutations were included, of which the plot samples included a total of 387 (78.5%). We used the chi-square test to evaluate the difference in gene mutation frequency in each group of samples. The results of the difference test for all genes are as follows.

### 2.8. Protein–Protein Interaction (PPI) Networks

PPI prediction analysis for the FDCSP-related proteins was performed using the STRING database version 11.0 (https://string-db.org/, accessed on 1 March 2022), which predicts interactions based on relevant experimental data.

### 2.9. BioGPS Database

BioGPS (http://biogps.org, accessed on 1 March 2022) is a free online extensible and customizable gene annotation portal tool that supplies a complete resource about gene and protein function. It displays a survey across diverse normal human tissues from the U133plus2 Affymetrix microarray. The values shown are z-scores produced by the barcode function of the R package “frma”. A z-score >5 suggests that the gene is expressed in that tissue.

### 2.10. Tissues Collection and Ethics Statements

A total of 20 untreated HNSC tumor specimens and matched normal tissues were obtained from Nanfang Hospital of Southern Medical University, Guangzhou, from 2021 to 2022. Of the 20 cases, there were 17 HPV+ and 3 HPV−. (Detected by QPCR) All patients were informed with written consent and the Ethics Committees of Nanfang Hospital approved the collection and use of all clinical specimens (NO: NFEC-2018-027).

### 2.11. RNA Isolation, Reverse Transcription, and qRT-PCR

Total RNA was extracted from the cells using Trizol (RNA Isolator (Vazyme Biotech Co., Ltd., Nanjing, China)). Reverse transcription (RT) and qPCR were performed in accordance with the manufacturer’s instructions (Vazyme Biotech Co., Ltd., Nanjing, China). RT-qPCR for each gene was repeated three times. Quantification amounts were normalized to GAPDH levels. Primers are listed in the Appendix A. HPV status was distinguished by CT values (CT > 35: negative, CT < 35: positive).

### 2.12. Immunoblot and Immunohistochemistry Assay

A total of 6 samples of collected oropharyngeal cancer specimens (3 HPV− and 3 HPV+ determined by qRT-PCR mentioned before) were used. FDCSP (Solarbio Ltd., Beijing, China,) and CXCL13 (SAB Ltd. Baltimore, MD, USA) antibodies were both diluted by 1:500.

## 3. Results

### 3.1. Identification of Differentially Expressed Genes

Seven GEO datasets from various nations were included in the analysis (Figure 1A). Gene expression levels of merged GEO series with batch effects adjusted that were standardized before and after processing are presented in Appendix A.

The combined datasets included 167 HPV+ HNSC samples and 574 HPV− HNSC samples. In total, 52 DEGs with |log2 FC| > 2.5 in the HPV+ HNSC samples that were compared with the HPV− HNSC samples were identified. There were 4 upregulated genes and 5 downregulated genes. The DEGs are presented in the heatmap plot and volcano plot (Figure 1B,C).

Based on the better prognosis of HPV+ HNSC, we intended to focus on which genes or proteins inhibited tumor progression, so we focused on genes that were highly expressed in HPV+ HNSC. Outcomes of upregulated genes in DEGs were estimated in TCGA-HNSC by TIMER2.0 (Figure 2, DEGs: Neurofilament Heavy Chain (NEFH) and FAM3 Metabolism Regulating Signaling Molecule B (FAM3B) and Keratin, Type I Cytoskeletal 19(KRT19)). The associations between gene expression and overall survival (OS) in TCGA-HNSC patients were estimated. The Kaplan–Meier plot analysis revealed that increased FDCSP, NEFH, and FAM3B expression was associated with better overall survival in HPV+ HNSC. Based on gene function and viral infection, we focused on FDCSP, a gene strongly associated with immune cells.

### 3.2. FDCSP Expression Levels in TCGA-HNSC and Normal Tissues

The expression levels of FDCSP in head and neck cancer patients with different HPV statuses from the TCGA HNSC database were analyzed by TIMER. FDCSP was found to be more abundant in TCGA-HNSC tissue than in normal tissue and to be more abundant in HPV+ HNSC than in HPV− HNSC (Figure 2A and Appendix A).

FDCSP is secreted from follicular dendritic cells (FDCs), which are dendritic cells (DCs). Because FDCs are characterized by high tissue selectivity, we reasoned that FDCSP would be the same. The FDCSP expression in human normal tissues was then identified using the BioGPS web tool. We obtained the raw data of FDCSP expression and filtered out the expression level >2 cell types in Figure 3B. FDCSP showed the highest expression levels in tonsil tissue and gingival epithelium, with expression levels larger than 5 (raw data were downloaded from BioGPS).

### 3.3. Immune Infiltration Analysis Related to FDCSP in HNSC

Since the source and function of FDCSP are closely related to the immune system, we attempted to investigate the relationship of FDCSP in HNSC immunity. We estimated stromal, immune, and ESTIMATE scores in GEO cohorts based on gene expression (Appendix A). We know that the FDCSP is closely related to the immune function of tumor tissue.

Using the TIMER2.0 database, we mapped the immune infiltration of HNSC in both TCGA and GEO cohorts (Appendix A, immune infiltration algorithm: CIBERSORT_ABS). FDCSP was observed to have a substantial positive correlation with B memory cells (R = 0.12), B plasma cells (R = 0.258), CD4+ memory resting cells (R = 0.141), and macrophage M0 cells (R = 0.15), and a significant negative correlation with CD4+ memory activated cells (R = −0.101). In HPV+ HNSC, FDCSP expression had a significant positive correlation with B cells (B memory cells: R = 0.428, B naive cells: R = 0.282, B plasma cells: R = 0.239), CD4+ memory resting cells (R = 0.374), regulatory T cells (Tregs, R = 0.511), macrophage M1 cells (R = 0.394), resting dendritic cells (DC) (R = 0.25), T gamma delta cells (R = 0.285) and T follicular helper cells (TFHs, R = 0.599) (Figure 3C). Immune cell infiltration was calculated by the CIBERSORT_ABS algorithm. The TIMER2.0 database estimation mode and immune cell infiltration of GEO cohorts in HNSC with two HPV statuses were calculated (Figure 3C).

To further clarify the effect of FDCSP on immune infiltration in HPV+ HNSCs, we measured the levels of various immune cells using the CIBERSORT score in the GEO cohort. The composition of 21 kinds of immune cells in HNSC GEO cohorts with or without HPV infection is shown in Appendix A. According to the expression level of FDCSP, the samples were divided into a high expression group (≥50%) and a low expression group (<50%), and the correlation analysis with the immune cell score was carried out (Figure 3D, *p* < 0.05 and |R| > 0.2 were filtered out). In HPV− HNSC, FDCSP was positively related to B memory cells (R = 0.287). In HPV+ HSNC, FDCSP was associated with more TFHs (R = 0.333), B memory cells (R = 0.337), CD8+ T cells (R = 0.229), CD4+ memory cells (R = 0.248), and macrophages (M0 R = 0.244, M1 R = 0.258).

HPV+ HNSC in the GEO cohort was divided into two groups according to the level of FDCSP expression, and the correlation between immune cells was assessed (Appendix A).

### 3.4. The Outcome Relations between the FDCSP and Immune Cell Types in HNSC

We further analyzed the prognosis of FDCSP and the degree of infiltration of immune cells in HNSC. From the TIMER2.0 database, we constructed a Cox model of FDCSP and immune cells in HNSC (Appendix A). In HPV− HNSC, FDCSP/CD4+ naive T cells showed an increased risk (Z score = 2.022), whereas FDCSP/TFHs showed a decreased risk (Z score = −1.076). In HPV+ HNSC, FDCSP and CD8+ T cells were associated with a reduced probability of HPV+ HNSC (Z score = −2.106). In Figure 4A, higher FDCSP and more CD8+ T cells were associated with a better prognosis in HPV+ HNSC (HR = 0.265). As illustrated in Figure 4B,D, Kaplan–Meier plots showed that lower FDCSP with more CD4+ naive T cells (HR = 2.23) or fewer TFHs (HR = 0.621) was associated with a worse prognosis in HPV− HSNC.

### 3.5. FDCSP Is Related to TP53 Mutation

In HNSC, the distribution of genetic mutations was investigated in groups according to high/low FDCSP expression. Significantly mutated gene mutation profiles presented TP53 (82.5%) and CASP8 (11.3%) in HNSCs (Figure 5).

We used TIMER2.0 to explore the correlation between FDCSP and TP53, and the TP53 expression level was positively correlated with FDCSP, regardless of HPV status (Figure 6A). The mutated TP53 group showed significantly less FDCSP than the wild-type TP53 group in HPV+ HNSC (*p* < 0.001). However, there was no significant difference in HPV− HNSC. (Figure 6B).

These findings may provide insight into the potential association between FDCSP expression and somatic variation, leading to potential immunological and prognostic features in HNSC.

In the GSE65858 cohort with TP53 mutation data, we investigated TP53 mutations according to FDCSP high/low expression and HPV status. Although the expression of FDCSP was significantly increased in mutated TP53 in HPV− HNSCs, there was no significant difference in HPV+ HNSCs (Appendix A).

### 3.6. Genes and Functions Related to FDCSP in HNSC

To explore the coexpression network related to FDCSP, the Limma R package was further constructed based on the transcriptional expression profile of FDCSP in HPV+ HSNC (using merged GEO cohorts). We intersected the DEGs in both datasets and obtained 159 genes (112 upregulated genes and 47 downregulated genes) significantly related to FDCSP in HPV+ HNSC (FDR < 0.01; fold change > 1.5).

Then, we used these 159 DEGs to carry out functional enrichment analysis (FDR < 0.1, *p* < 0.05). Gene Oncology (GO) analysis results are listed in Figure 7. As shown, DEGs were primarily enriched mostly in immune regulation and cell activation (leukocytes, lymphocytes, T cells) (Figure 7A). Regarding biological processes (GOBP), DEGs were mainly associated with immune system processes (Figure 7B). Multiple cellular components (GOCP) were enriched by DEGs, including the extracellular region and membrane (Figure 7C). Moreover, a variety of molecular functions (GOMF) were enriched by DEGs, including identical protein binding, signaling receptor binding, and chemokine signaling pathways (CCR chemokine receptor binding, chemokine receptor binding, and CCR10 chemokine receptor binding) (Figure 7D). The Kyoto Encyclopedia of Genes and Genomes (KEGG) pathway enrichment analysis results showed that FDCSP-DEGs were significantly enriched in pathways, such as cytokine–cytokine receptor interaction, the chemokine signaling pathway, and Th17 cell differentiation (Figure 7E).

To determine the underlying mechanism of FDCSP in HNSC, we carried out gene set enrichment analysis (GSEA) to identify FDCSP-related pathways in HPV+ HNSC using the KEGG dataset. The top ten enrichment scores (ES) are shown in Figure 6E. The DEGs were enriched in immune-related pathways, such as antigen processing (ES = 0.6650), the T-cell receptor signaling pathway (ES = 0.5959), viral myocarditis (ES = 0.7527) and the chemokine signaling pathway (ES = 0.5532) (Figure 6F FDR < 0.25). These findings suggested that FDCSP potentially facilitates chemokine pathways in HPV+ HNSC.

### 3.7. FDCSP Correlates with CXCL13 in HPV+ HNSC

In the previous functional enrichment analysis, we found that FDCSP is closely related to the chemokine pathway in HPV+ HNSC. We then intersected the DEGs of FDCSP in HPV+ HSNC and constructed a protein–protein interaction (PPI) network (Figure 8B). We found that CXCL13 was positively correlated with HNSC in the GEO cohorts (Figure 8C). Likewise, CXCL13 was expressed differently in HPV+/− HNSC (Appendix A).

In HNSC tissues collected from Nanfang Hospital (n = 20), there were 17 HPV+ and 3 HPV− samples. We detected mRNA expression of FDCSP and CXCL13. FDCSP was highly expressed in HPV+ and was related to a worse prognosis (Figure 8C,E). In either 3 samples of oropharyngeal cancer tissues collected, the protein expression levels of FDCSP and CXCL13 were both higher in HPV+ cancer tissues (Figure 8D).

## 4. Discussion

HPV-induced HNSC tumorigenesis has been demonstrated to exhibit a distinct pathological pattern. Despite poor overall survival in HNSC, HPV+ HNSC showed a better prognosis than HPV− HNSC. Recent studies have shown that improved outcomes in patients with HPV+ HNSC are associated with the high responsiveness of these tumors to chemotherapy, radiotherapy, and targeted therapy [24,25]. Immunotherapy is defined as the treatment of utilizing the immune system of the body to destroy cancer cells. Immune cell infiltration plays a critical role in tumor development, metastasis, and invasion. In the tumor microenvironment, dynamic changes in immune infiltration may act as a double-edged sword.

To determine which gene contributes to immune dynamics in HPV+ HNSCs, we analyzed seven GEO cohorts and identified DEGs. Four upregulated DEGs were filtered out. Among them, FDCSP was expressed at significantly higher levels in HPV+ HNSC and was associated with a favorable prognosis in HPV+ HNSC. FDCSP is specifically highly expressed in the gingival epithelium, tonsils, and tonsil epithelium. FDCSP is a unique secreted protein by follicular dendritic cells (FDCs) that has received little attention and research. FDCs contribute to the pathogenesis of a variety of immune-related diseases, including HIV/AIDS, prion diseases, chronic inflammation, and autoimmune diseases [26,27,28]. However, no research about FDCs on the occurrence and development of HPV virus and tumors has been discovered. TNF-α and IL-1β stimulate human FDCSP gene transcription by targeting YY1, GATA, C/EBP2, and C/EBP3 in the FDCSP gene promoter [29]. FDCSP can regulate IgA production [30,31,32]. FDCSP is also expressed in the periodontal ligament [33], and its abnormally high expression promotes the formation of osteoclasts [34,35]. When combined with the fact that the expression of FDCSP is increased in TCGA-HNSC in HPV+ and the overall survival rate is improved, we have reason to believe that FDCSP plays a critical role in HPV+ HNSC. As a result, we believe that studying FDCSP in HNSC in relation to HPV is critical for understanding the dynamics of HNSC immunity.

We screened out FDCSP using HPV+ versus HPV− in multiple GEO datasets and validated them in the TCGA database. Additionally, increased FDCSP expression indicates a better prognosis in HNSC. We further calculated the immune score based on the GEO cohort’s gene expression matrix and performed correlation analysis with FDCSP. FDCSP was significantly associated with the Immune Score and ESTIMATE Score in HPV+ HNSC. The CIBERSORT algorithm is an algorithm that calculates the scores of 21 immune cells in the bulk expression matrix based on the model obtained by single-cell sequencing [36]. We further performed calculations of immune cell infiltration in tumors in the TCGA cohort and the GEO cohorts using the CIBERSORT algorithm. We calculated the correlation of FDCSP with immune cell infiltration in HPV+ and HPV− HNSC. Although these immune cell types significantly associated with FDCSP were not identical in the two datasets, we were able to find some shared characteristics. Although FDCSP is secreted by FDCs, we found no correlation between FDCSP expression and DC cell score, whereas the CIBERSORT algorithm excluded the FDC immune cell score. FDCSP was predominantly positively associated with B cells in HPV− HNSC. FDCSP was mainly associated with TFHs in HPV+ HNSC.

Lower FDCSP levels with lower TFH infiltration had a significantly worse prognosis in the TCGA HPV− HNSC group than those with higher TFH infiltration. In TCGA HPV+ HNSC, higher FDCSP levels with higher TFH infiltration have a significantly better prognosis than those with lower TFHs. These findings suggest that FDCSP works in tandem with TFHs in HPV+ HNSC. Follicular T helper cells, as the name implies, are a subset of activated T cells that specialize in providing T-cell assistance to corresponding B cells in germinal centers (GCs). GCs are specialized microenvironments formed after infection where activated B cells can mutate their B-cell receptors to undergo affinity maturation [37]. TFHs are essential for GC B-cell maturation. TFHs are commonly found at the periphery of B-cell follicles and constitutively express the B-cell follicle homing receptor C-X-C chemokine receptor type 5 (CXCR5) [38]. These T-B interactions are critical in supporting isotype switching and affinity maturation of B cells. TFHs have been found to be closely related to longer progression-free survival in HNSC patients [39] and to prognosis in breast cancer [40], lung cancer [41], and endometrial cancer [42], which provides a valuable resource for the investigation of immunotherapy for HPV+ HNSC. The single-cell RNA sequencing for the immune landscape of HPV+/− HNSC by Anthony found that there are fewer TFHs in HPV− HNSC and concluded that T follicular helper signature was associated with favorable survival in TCGA patients [39]. A recent study has highlighted the role of TFHs in vaccine design [43].

It is now clear that the p53 status of cancer cells has a significant impact on the immune response [44]. HNSCs have a high prevalence of TP53 mutations [45], and TP53 mutations are associated with prognosis in HPV+ tumors. Thus, using the data of tumor mutations in TCGA, we discovered that in HPV+ HNSC, the expression of FDCSP was significantly positively correlated with TP53 (R = 0.444). In addition, TCGA HPV+ HNSC was divided into two groups based on the presence or absence of TP53 mutation. FDCSP levels were significantly higher in the wild-type (WT) TP53 gene than in the mutated TP53 gene. We hypothesize that in the presence of a TP53 mutation, FDCSP expression would be downregulated. We speculate that the expression of FDCSP would be downregulated in the context of TP53 mutation. Alternatively, we could hypothesize that in the genetic context of TP53 mutations, there would be a corresponding decrease in FDC cells.

In HPV+ HNSC, aberrantly expressed genes were enriched in immune processes, including the chemokine signaling pathway, T-cell activation, and cytokine signaling pathway [46]. To further elucidate the role of FDCSP in HNSC, we screened GEO cohorts for differentially expressed genes associated with FDCSP. After classifying differential genes into functional clusters using GSEA and GO/KEGG enrichment, FDCSP function was concentrated in the immune process, where the activation of T cells and the activation of the chemokine family were significantly correlated. Combined with the PPI network, we finally related FDCSP with CXCL13. Chemokine (C-X-C motif) ligand 13 (CXCL13), also known as B lymphocyte chemoattractant (BLC) or B-cell-attracting chemokine 1 (BCA-1), is a protein ligand that in humans is encoded by the CXCL13 gene [47,48]. It is worth mentioning that follicular dendritic cell sarcoma (FDCS), a disease characterized by the malignant proliferation of FDCs itself, exhibits overexpressed FDCSP and overexpressed CXCL13, which cooperate with other genes to diagnose FDCS [49,50]. Notably, FDC-secreted CXCL13 attracts not only B cells but also T follicular cells via CXCR5 [51].

In normal lymphoid tissue, FDC networks can regulate recirculating resting B cells and divert antigen-activated B cells to clonal expansion to generate germinal centers (GCs) [52]. FDCs secrete CXCL13, a B-cell chemokine, and GC B cells express the CXCL13-binding chemokine receptor CXCR5 [28]. Although the FDCSP protein itself has low chemotaxis, FDCSP was found to enhance the chemotaxis of CXC chemokines, and the acute stimulation of FDCSP could cooperate with chemokines to promote the migration of B cells [53]. In GEO cohorts, the expression of FDCSP and CXCL13 in HNSC was positively correlated regardless of HPV status. However, CXCL13, like FDCSP, was significantly highly expressed in HNSCs (Appendix A) and even more upregulated in HPV+ HNSC. As previously stated, the FDCSP gene is located on chromosome 4q13, which is adjacent to clusters of C-X-C chemokines. The mechanism by which HPV infection results in an increase in FDCSP and CXCL13 expression remains unknown.

## 5. Conclusions

We have demonstrated for the first time that FDCSP is required for the immune infiltration of HNSC. Increased FDCSP was significantly associated with a better prognosis, especially in HPV+ HNSC. Due to their close relationship with CXCL13, FDCs appear to play an important role in modifying the immune environment, particularly THFs, during the HPV+ HNSC tumor immunity process. Additional single-cell sequencing and experiments are necessary to decipher the dynamic role of FDCs in the immune response of HPV-induced tumors as well as the causal relationship between FDCs and CXCL13.

## Figures and Tables

**Figure 1 biomolecules-12-01458-f001:**
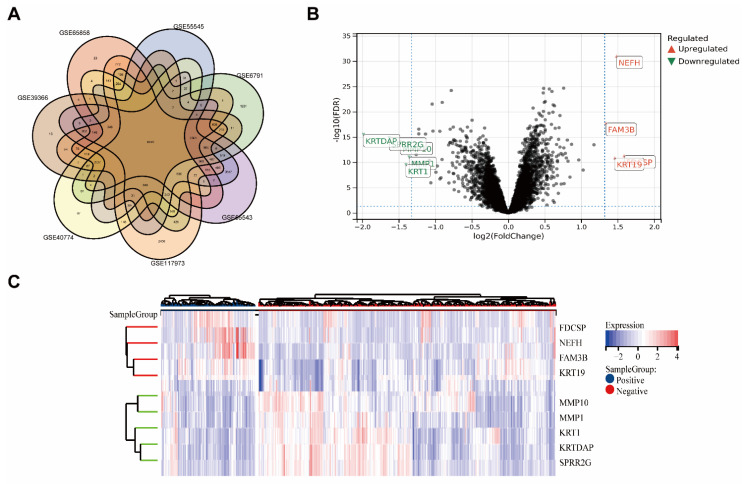
Screening of HPV-related differential genes using the GEO database. (**A**) Venn diagram for GEO datasets. (**B**) Volcano map of differentially expressed genes in HPV+ verse HPV− group. (FDR < 0.05, fold change = 2). (**C**) Heatmap of differentially expressed genes, clustered by HPV status.

**Figure 2 biomolecules-12-01458-f002:**
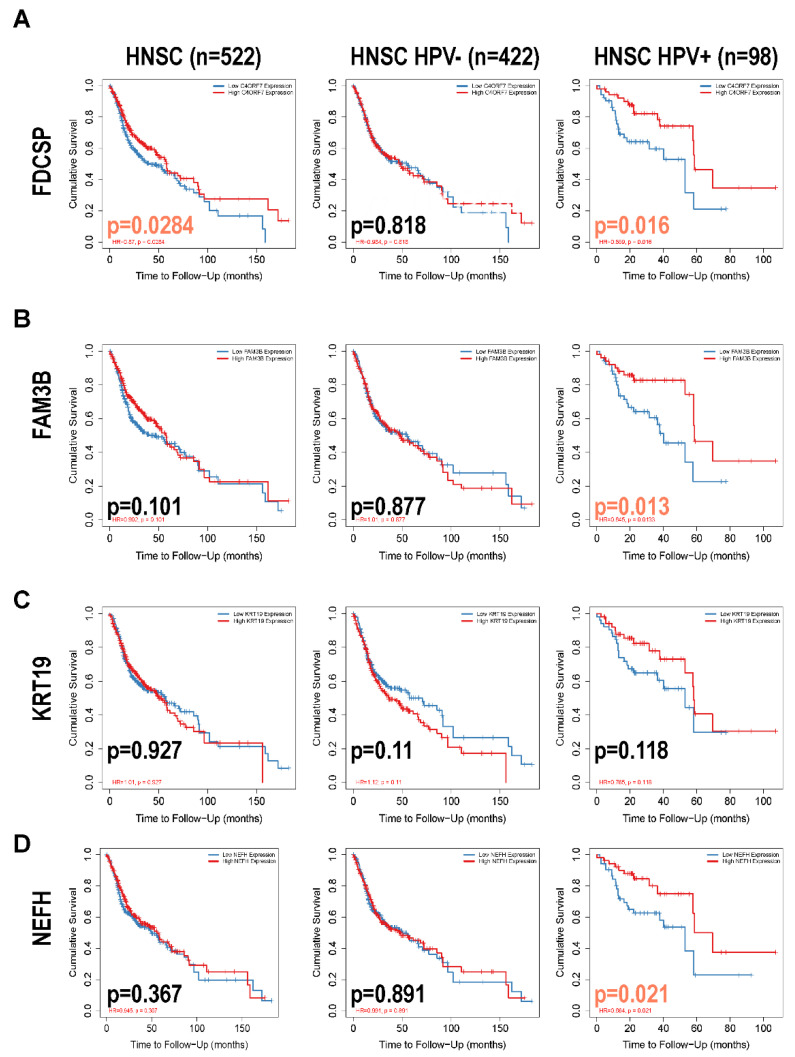
Overall survival analysis of DEGs in HPV status in HNSC. (**A**–**D**): Kaplan–Meier curves associated with upregulated differential genes (FDCSP, FAM3B, KRT19, NEFH) grouped by HPV status.

**Figure 3 biomolecules-12-01458-f003:**
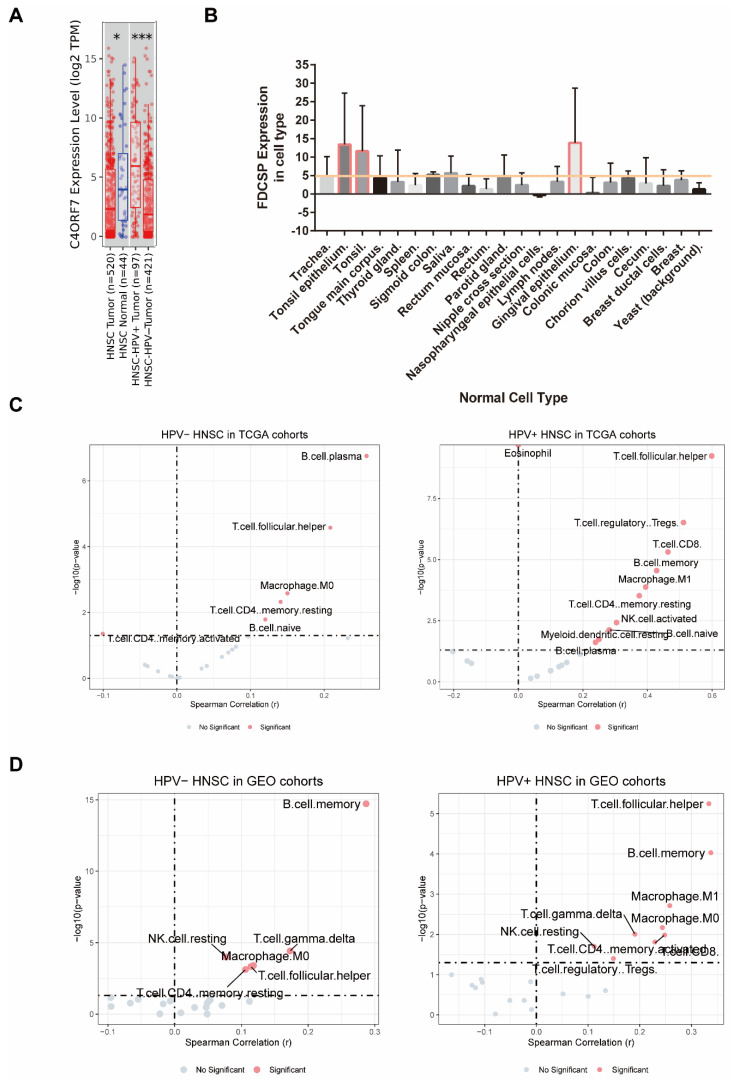
The relationship between FDCSP and immune cells in HNSC. (**A**) Expression of FDCSP in HSNC (from Timer2.0 database, grouped by HPV status, A z-score >5 suggests that the gene is expressed in that tissue). (**B**) FDCSP expression in human normal tissues. (**C**) Correlation between FDCSP expression and immune cell types in TCGA-HNSC cohorts (CIBERSORT_ABS). (**D**) Correlation between FDCSP expression and immune cell types of HNSC in GEO cohorts (CIBERSORT_ABS). (*x*-axis: Spearman Correlation(r), *y*-axis: -log10(p), Dotted line as dividing line: x at 0 and y at 1.301). *: *p* value<0.05, ***: *p* value<0.005.

**Figure 4 biomolecules-12-01458-f004:**
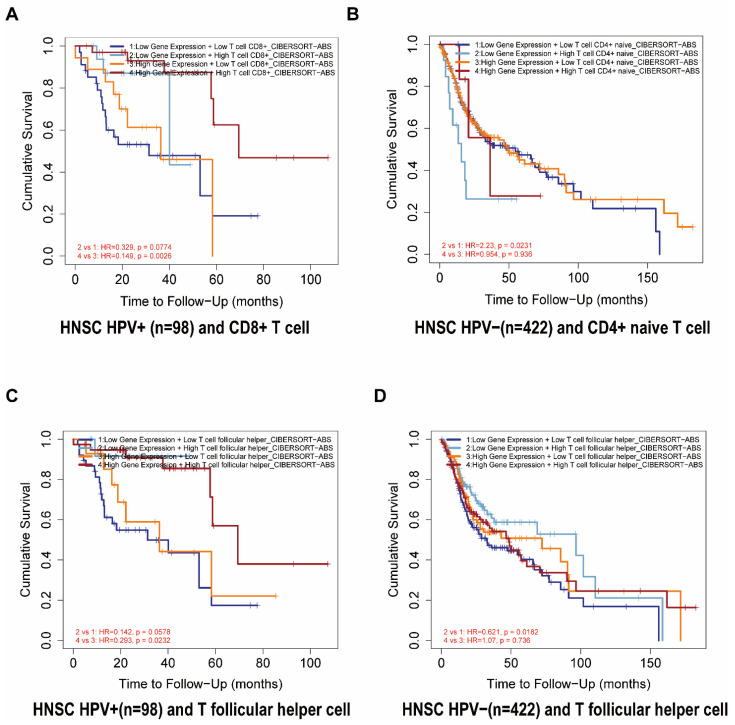
The relationship between FDCSP and immune cells in TCGA-HNSC. (**A**–**D**) Kaplan–Meier curves for the corresponding immune infiltrates and FDCSP expression in HPV status. ((**A**) FDCSP in HPV+ with CD8 + T cells; (**B**) FDCSP in HPV− with CD4 + T cells; (**C**) FDCSP in HPV+ with TFHs; (**D**) FDCSP in HPV− with TFHs).

**Figure 5 biomolecules-12-01458-f005:**
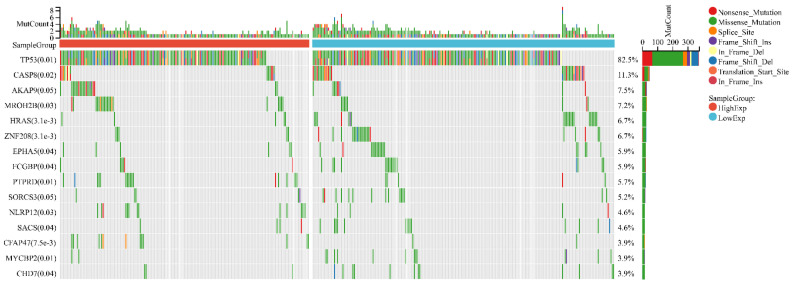
Variation in gene mutation frequency with FDCSP in HNSC.

**Figure 6 biomolecules-12-01458-f006:**
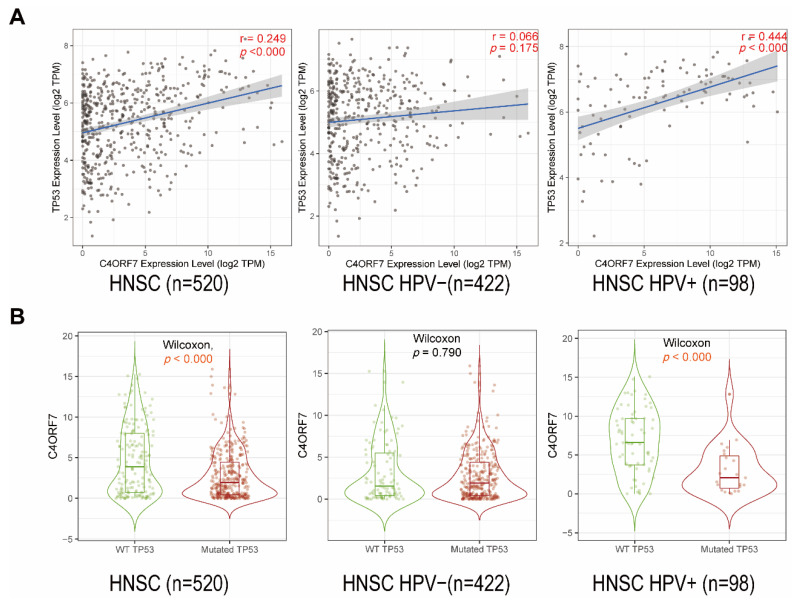
Correlation between FDCSP and TP53 in HNSC. (**A**) Correlation between FDCSP and TP53 expression (Spearman). (**B**) Violin plot and boxplot showed correlation of FDCSP with mutated/wild-type (WT) TP53, Wilcoxon algorithm was used for calculating statistical difference.

**Figure 7 biomolecules-12-01458-f007:**
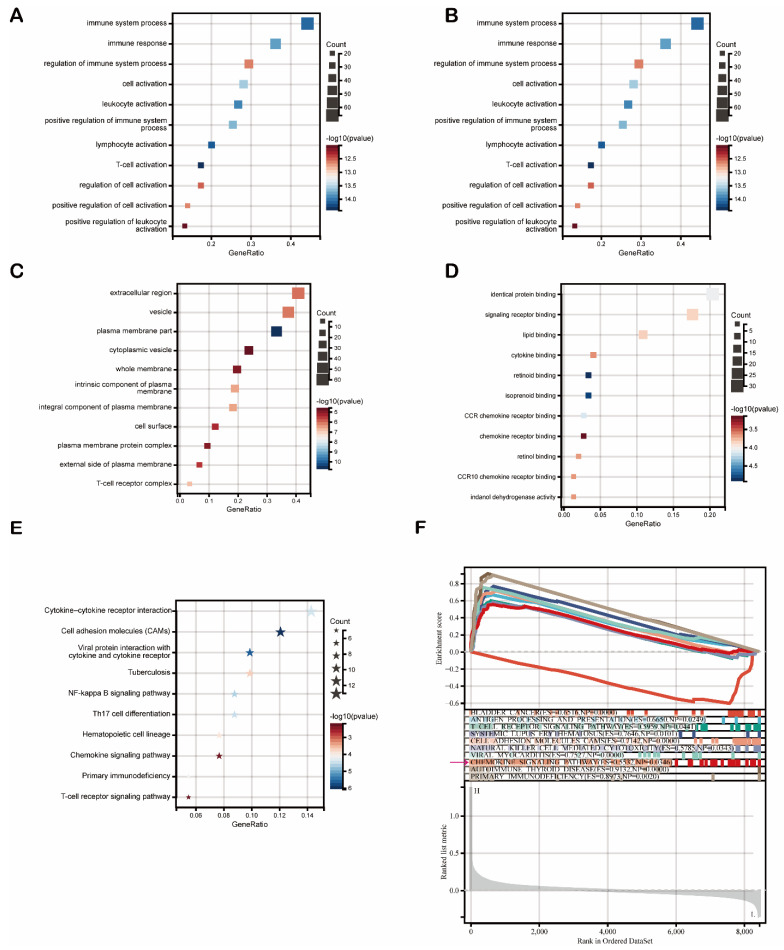
Enrichment analysis of FDCSP. (**A**) GO enrichment. (**B**) GO biological processes. (**C**) GO cellular components. (**D**) GO molecular functions. (**E**) KEGG. (**F**) GSEA of KEGG.

**Figure 8 biomolecules-12-01458-f008:**
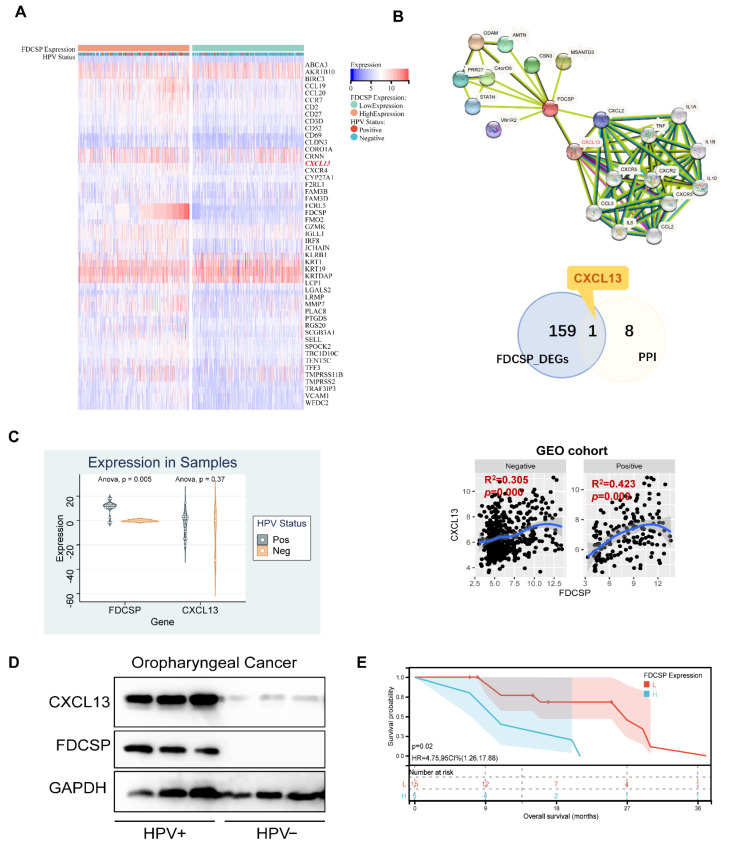
Analysis of FDCSP-related genes. (**A**) Heatmap of differential genes in the high/low FDCSP group in GEO-HNSCs (50/159 DEGs is shown). (**B**) String PPI network and Venn diagram of DEGs and PPIs. Correlation of FDCSP and CXCL13 in HPV+/− HNSC in GEO cohorts. (**C**) FDCSP and CXCL13 mRNA expression in oropharyngeal cancer samples versus paired normal tissues. (**D**) FDCSP and CXCL13 protein expression in oropharyngeal cancer samples (3 HPV+ and 3 HPV− samples). (**E**) FDCSP mRNA expression relates with prognosis in oropharyngeal cancer samples (low group/L: FDCSP expression less than the median and high group/H: FDCSP expression more than the median).

## Data Availability

Not applicable.

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
