# Peer review of "FDCSP Is an Immune-Associated Prognostic Biomarker in HPV-Positive Head and Neck Squamous Carcinoma"

_biomolecules, 2022, doi:10.3390/biom12101458_

Round 1
Reviewer 1 Report
This is an interesting manuscript that is based on the analyses of existing databases. In addition to the quantitative RT-PCR, the manuscript will be significantly enriched if the authors validate their results by performing additional analyses in other head and head squamous cell carcinoma.
For instance, it would be interesting to determine p53 levels by immuno-histochemistry and correlate its levels with DNA mutations. Overall, the manuscript deserves publication.
Author Response
Comments
This is an interesting manuscript that is based on the analyses of existing databases. In addition to the quantitative RT-PCR, the manuscript will be significantly
enriched if the authors validate their results by performing additional analyses in other head and head squamous cell carcinoma.
For instance, it would be interesting to determine p53 levels by immuno-histochemistry and correlate its levels with DNA mutations. Overall, the manuscript deserves publication.
Response:
We would like to thank you for your careful reading, helpful comments, and constructive suggestions, which has significantly improved the presentation of our manuscript. We have carefully considered all comments from the reviewers and revised our manuscript accordingly. The manuscript has also been double-checked, and the typos and grammar errors we found have been corrected.
We collected protein samples from 6 oropharyngeal carcinomas and performed protein level detection. The results are shown in Figure 7D, and the original data are attached for reference.
It's interesting to note that TP53 mutations have also been reported to the rare FDC malignant proliferative disease follicular dendritic cell sarcoma (FDCS). Thank you for your guidance on the next research direction. We will focus on the correlation between TP53 expression, mutant and FDCSP expression in our next work.
Reviewer 2 Report
In this study Wu et al., analyzed 7 datasets from the gene Expression Omnibus (GEO) database. The datasets contain samples of HPV- and HPV+ HNSCs. Bioinformatic analysis identified differentially expressed genes in HPV+ HNSCs in comparison to HPV- HNSCs with four up-regulated genes containing follicular dendritic cell secreted protein (FDCSP). High expression of FDCSP in HPV+ HNSCs was associated with better cumulative survival and infiltrating T follicular helper cells (TFHs) and B memory cells and the combination of high FDCSP expression and high amounts of CD8+ T cells or TFHs was associated with better prognosis in HPV+ HNSCs. FDCSP expression was positively correlated with TP53 expression regardless of the HPV status and HPV+ HNSCs with mutated TP53 showed significantly less FDCSP expression. Finally, in protein-protein-interaction network FDCSP and CXCL13 expression was positively correlated in HNSCs.
At this stage, the authors provided bioinformatics analysis of datasets from others groups with the hypothesis that FDCSP expression is involved in immune cell infiltration of HPV+ HNSCs and thereby affect prognosis. The authors speculated that FDCSP expression and TFHs works in tandem in HPV+ HNSCs. However, validation of their findings in tumor samples on protein level are missing. Detection of, for example, FSCSP or CXCL13 expression in immunohistochemistry or the presence of follicular dendritic cells (FDCs) or TFHs and subsequent correlation analysis would support their hypothesis and strengthen the relevance of their findings, especially since there are no information about the role of FDCs and less information (only one citation) about TFHs in HNSCs provided in the introduction or discussion section.
Further comments:
- Figure 2: FAM3B and NEFH were up-regulated in HPV+ HNSCs, too. A high FAM3B and NEFH expression also significantly affect survival, but is not mentioned in the text. Please comment.
- Figure 4: labeling of the figure (Kaplan-Meier curves) is hard to follow and should arranged in a proper way to see all details and colors. Labeling of Figure 4 C and D (HPV+ and HPV-) is inverted in the legends (lane 254).
In general, results should be written more precisely. Some figures (for example Figure 1B, 1C) are mentioned in the text or figures are not correctly listed in the text (paragraph 3.6, should describe figure 7).
- Figure legends should provide more details.
Author Response
We would like to thank you for your careful reading, helpful comments, and constructive suggestions, which has significantly improved the presentation of our manuscript.
Comments
However, validation of their findings in tumor samples on protein level are missing. Detection of, for example, FSCSP or CXCL13 expression in immunohistochemistry or the presence of follicular dendritic cells (FDCs) or TFHs and subsequent correlation analysis would support their hypothesis and strengthen the relevance of their findings, especially since there are no information about the role of FDCs and less information (only one citation) about TFHs in HNSCs provided in the introduction or discussion section.
Response:
We performed protein level detection on six oropharyngeal carcinoma protein samples. Figure 7D presents the results, and the attached original data are included. However, due to the size of the collected samples, we were unable to obtain ideal immunohistochemical specimens for tissue-level validation. Unfortunately, there is no information regarding HPV status for the pertinent histochemical specimens that we attempted to locate on The Human Protein Atlas (https://www.proteinatlas.org/ENSG00000181617-FDCSP/pathology/head+ and+neck+cancer#ihc).
We added the missing citations in lanes 393–395 as a result of your comments. This study discovered that TFHs were highly expressed in HPV+ HNSCs and correlated with a better prognosis at the single-cell level, but FDCs were left out. Unfortunately, we are unable to conduct additional analysis of this portion of the public data due to the lack of equipment for single-cell RNA sequencing analysis in the research team.
In addition to protein samples, we also examined the literature and discovered that CXCL13 was positively correlated with FDCSP in Follicular Dendritic Cell Sarcoma (FDCS), further supporting the significant correlation between FDCSP and CXCL13 (Lanes 419-423).
Further comments
Point1: Figure 2: FAM3B and NEFH were up-regulated in HPV+ HNSCs, too. A high FAM3B and NEFH expression also significantly affect survival but is not mentioned in the text. Please comment.
Response:
The filter of core molecules is based on the function of the molecules as well, considering that HPV+ HNSC is in a state of viral infection. A molecule associated with metabolism is called FAM3B, and a biomarker of neuronal damage is called NEFH (Neurofilament Heavy Chain). We concentrated on FDCSP among the three genes significantly associated with survival based on the current results. The results now include this section (Lane 182-188).
Point2:Figure 4: labeling of the figure (Kaplan-Meier curves) is hard to follow and should arranged in a proper way to see all details and colors. Labeling of Figure 4 C and D (HPV+ and HPV-) is inverted in the legends (lane 254). In general, results should be written more precisely. Some figures (for example Figure 1B, 1C) are mentioned in the text or figures are not correctly listed in the text (paragraph 3.6, should describe figure 7).
Response:
In Figure 4, an inversion error has been corrected, and the labels in the K-M diagram have been moved and given a different color to make them easier to understand.
Point3: - Figure legends should provide more details.
Response: The figure legends have been revised and improved further.
Round 2
Reviewer 1 Report
No additional comments, publication is recommended for this manuscript
Reviewer 2 Report
The authors address all questions and comments.